

# Starvation and re-feeding of Gilthead seabream (*Sparus aurata*) and European seabass (*Dicentrarchus labrax*) co-cultured with glasswort (*Salicornia europaea*) in a polyculture aquaponic system

Ioannis Mitsopoulos[1], Iliana Gesthimani Kontou[1], Konstantinos Babouklis[1], Nikolaos Vlahos[1,2], Panagiotis Berillis[1], Efi Levizou[3] and Eleni Mente[1,4]

[1] Department of Ichthyology and Aquatic Environment, School of Agricultural Sciences, University of Thessaly, Volos, Greece
[2] Department of Fisheries and Aquaculture, School of Agricultural Sciences, University of Patras, Mesolonghi, Greece
[3] Department of Agriculture Crop Production and Rural Environment, School of Agricultural Sciences, University of Thessaly, Volos, Greece
[4] School of Veterinary Medicine, Laboratory of Ichthyology- Culture and Pathology of Aquatic Animals, Aristotle University of Thessaloniki, Thessaloniki, Greece

Corresponding authors
Nikolaos Vlahos, vlachosn@upatras.gr
Panagiotis Berillis, pveril@uth.gr

## ABSTRACT

The aim of this study was to evaluate the effect of starvation and refeeding on the growth and food intake of gilthead seabream (*Sparus aurata*) and seabass (*Dicentrarchus labrax*) and on the growth and nitrogen uptake of glasswort (*Salicornia europaea*) in a polyculture aquaponic system under 12 ppt salinity for 75 days. Nine small-scale autonomous aquaponic systems were used, each containing 10 gilthead seabreams (average weight of 6.33 ± 0.73 g and average length of 5.73 ± 0.72 cm) and 10 seabasses (5.82 ± 0.77 g and 6.35 ± 0.45 cm), as well as five glasswort plants. Three fish feeding treatments were performed, a control (A), in which fish were fed daily until satiation, and two fasting treatments for 4 (B) and 7 days (C). Fish growth performance was significantly lower ($p < 0.05$) in the C treatment for both species compared to treatments A and B. Food consumption (FC) and feed conversion ratio (FCR) were significantly higher ($p < 0.05$) in treatment C. Glasswort growth performance was significantly higher in treatment C ($p < 0.05$). The results showed that the 4-day food-deprived fish were similar to the control fish by achieving partial compensatory growth. The more extended fasting period (7 days) resulted in significantly lower growth performance. The lipid and nitrogen retention levels in both species were significantly lower in food-deprived fish than in the control fish both before and during compensatory growth. The results suggest that a feeding schedule involving starvation–refeeding cycles is a promising feed management option for these species in polyculture aquaponic systems. The effect of food deprivation was also significantly beneficial ($p < 0.05$) for the growth performance of glasswort compared to the control treatment.

## INTRODUCTION

Aquaponics is a sustainable and eco-friendly food production method that combines aquaculture and hydroponic plant cultivation in a soilless recirculating system (*Somerville et al., 2014*). It promotes a solution to climate change, drinking water shortage, soil fertility loss, and other environmental impacts of aquaculture (*Buzby & Lin, 2014*; *Somerville et al., 2014*). In aquaponics, nitrifying bacteria oxidize the toxic ammonia produced by fish feces and unutilized feed and excrete urea as vital and usable nitrate for plants (*Cebron & Garnier, 2005*). Plant roots purify the water by absorbing nitrate, and water is then returned clean to fish tanks. In aquaponics, both fresh and brackish water can be used (*Vlahos et al., 2023a*; *Vlahos et al., 2023b*). In brackish water aquaponic systems halophytes, such as rock samphire and glasswort, are very important (*Vlahos et al., 2023a*; *Vlahos et al., 2023b*; *Pinheiro et al., 2020*), while Mediterranean euryhaline fish species, such as sea bream or sea bass can be used (*Stathopoulou et al., 2021*).

The impact of climate change is posing new challenges to the agricultural sector. Alternative production forms, such as aquaponics, are thus necessary (*Salam, Asadujjaman & Rahman, 2013*). Autonomous polyculture aquaponic systems, such as the Self-sufficient Integrated Multitrophic AquaPonic (SIMTAP) system, aim to reduce the dissolved nitrogenous products produced by aquatic organisms. In general, aquaponic polyculture is characterized as combining ecological principles with sustainability. According to *Allsopp, De Lange & Veldtman (2008)*, polyculture has environmental, social, and economic benefits. The function of the SIMTAP depends on the selection and placement of the plant responsible for the absorption of inorganic substances (nitrogen and phosphate) (*Chopin et al., 2001*).

*Salicornia europaea* is an annual succulent halophyte of the family Amaranthaceae (*Orlovsky et al., 2016*). It is considered one of the most salt-tolerant plant species in the world (*Ungar, 1987*; *Fan et al., 2011*). According to *Beeftink (1985)*, halophytes are significant members of the vegetation in salt marshes and marine wetlands, especially in areas with factors such as tides, salinity, soil moisture, and soil nutrients that favour perennial vegetation. Glasswort seeds remain dormant at high salinities and germinate only when salinity levels decrease, mainly in spring or during periods of heavy rainfall (*Waisel, 1972*). Glasswort can effectively remove nutrients, providing solutions in eutrophic environments, even in areas with increased salinity (*Brown et al., 1999*). It is a plant with considerable economic interest. It can be used as an edible plant, as a raw material for producing alcoholic beverages, and as a promising candidate plant for life support systems in space stations (*Ushakova et al., 2006*; *Tikhomirova et al., 2005*).

Compensatory growth is characterized by accelerated growth after a fasting period. This process reduces food and labor costs and minimizes feeding errors or overproduction (*Krogdahl & Bakke-McKellep, 2005*). During compensatory growth, the growth of

previously fasting fish is similar to that of fish that experienced regular feeding throughout the period (*Ali, Nicieza & Wootton, 2003*). Compensatory growth uses either total or partial deprivation to induce growth (*Eroldoğan et al., 2006a*; *Eroldoğan, Kumlu & Sezer, 2006b*). Feed deprivation periods can also be used to improve fish quality by avoiding excessive lipid accumulation (*Grigorakis & Alexis, 2005*). Compensatory studies have been done on many fish species, not always with the desired results (*Tunçelli & Pirhonen, 2021*).

Seabass (*Dicentrarchus labrax*) and seabream (*Sparus aurata*) are two of Europe's most important cultured fish, particularly for the Mediterranean region. In 2021, European production was 174,500 tons, with 81,369 tons of seabass and 93,131 tons of gilthead seabream produced (*FAO, 2021*). In Greece, the 2021 production of seabream and seabass amounted to 125,550 tons (73,050 tons of gilthead seabream and 52,500 tons of seabass) (*Federation of Greek Maricultures, 2020*). Seabass and seabream are ideal for brackish water aquaponics, as they are euryhaline species and can tolerate a wide range of salinities (*Rubio, Sánchez-Vázquez & Madrid, 2005*; *Stathopoulou et al., 2018*; *Vlahos et al., 2023a*). *Rossi et al. (2021)* reported that Mediterranean fish species, especially seabream, have satisfactory growth when reared in a polyculture system.

There is a lack of studies on polyculture brackish water aquaponic systems using different fish species , especially euryhaline species such as gilthead seabream and seabass. While research has shown that starvation and refeeding of gilthead seabream and seabass improves fish fillet quality, there is a gap of knowledge on the effect of starvation and refeeding in a polyculture aquaponic systems. The present study aims to bridge this knowledge gap and to investigate whether compensatory fish growth affects glasswort performance in a polyculture aquaponic system. Therefore, a polyculture brackish water aquaponic system with gilthead seabream (*Sparus aurata*) and seabass (*Dicentrarchus labrax*) co-cultured with glasswort (*Salicornia europaea*) was used.

## MATERIALS AND METHODS

### Experimental design—animal and plant procurement

The present study was conducted at the Laboratory of Aquaculture, Aquaponics Section at the Department of Ichthyology and Aquatic Environment, School of Agricultural Sciences, University of Thessaly in Greece. All experimental procedures were performed according to the guidelines of the EU Directive 2010/63/EU regarding the protection of animals used for scientific purposes and were applied by FELASA-accredited scientists (functions A–D). The experimental protocol was approved by the Ethics Committee of the Region of Thessaly, Veterinary Directorate, Department of Animal Protection-Medicines-Veterinary applications (n. 112841/23-03-2022). The experiment was conducted at the registered experimental facility (EL-43BIO/exp-01) of the Laboratory of Aquaculture, Department of Ichthyology and Aquatic Environment, University of Thessaly.

The rearing system consisted of nine small-scale autonomous aquaponic systems, functioning as integrated polyculture aquaponic systems where gilthead seabream, seabass (initial fish density 2.24 kg/m³), and glasswort were co-cultured for 75 days under 12 ppt salinity. Fish were provided to the laboratory by a local nursery facility (PhilosoFish

SA, Tapies, East Greece) and were divided into three 180-L aquariums for 30 days of acclimatization. In total, 90 individuals of gilthead seabream with average body weight and length of 6.33 ± 0.73 g and 5.73 ± 0.72 cm, respectively, and 90 individuals of seabass with average body weight and average body length of 5.82 ± 0.77 g and 6.35 ± 0.45 cm, respectively, were used. Ten individuals of each species were placed in each aquaponic system. Both species were placed and co-cultured in the same tank in cages (30 × 30 × 30 cm) to grow together, avoiding cannibalism.

During the adaptation period, gilthead seabream and seabass were adapted to a protocol of salinity reduction by 3–5 units every five days for 30 days until the salinity reached 12 ppt (*Stathopoulou et al., 2018*; *Rossi et al., 2021*; *Thomas et al., 2021*; *Vlahos et al., 2023a*). During this period, the fish were fed by hand with a 1.8-mm commercial diet (BIOMAR SA, GREECE).

Glasswort plants were collected from the Evros River Delta (Greece), transported to the laboratory, and adapted for 30 days to a salinity increase until 12 ppt (*Thomas et al., 2021*).

### Rearing system and conditions

The rearing system consisted of nine autonomous aquaponic recirculation systems with a total volume of 135 L that were kept in the laboratory at constant humidity (70%) and temperature (22 °C) throughout the rearing period. The experimental design was completely randomized, with three replicates per treatment, while the factor tested was feed intake (*Peres, Santos & Oliva-Teles, 2011*).

Each aquaponic system used in the present study was previously described by *Vlahos et al. (2023a)* and *Tsoumalakou et al. (2023)* and consisted of one 54-L glass aquarium (dimensions: 60 × 30 × 30 cm), one 54-L glass hydroponic subsystem (dimensions: 60 × 30 × 30 cm, cultivation area 1,800 cm$^2$) for the plants, and one sub-biofilter of 27 L. Furthermore, each cage was half divided by a 2.5 mm mesh. It was perforated in each fish tank to add gilthead seabream individuals to conserve them from seabass interspecies competition (Fig. 1).

The water flow rate was designed to be flowing through gravity (*Somerville et al., 2014*) from the hydroponic cultivation tank (located at the highest point of the system) to the fish-rearing tank (located at the intermediate point of the system) and ended up in the sump-bio filter. Moreover, the sump filter was designed to be upflow and downflow and was divided into three parts: the mechanical, biological, and pump sections. The mechanical filter consisted of a porous sponge inside a basket of one cm mesh, creating a surface area of 455 cm$^2$. It was used to capture fish feces and uneaten food. The biofilter's filter media had a surface area of 429 cm$^2$ and consisted of 3-L bio balls (AquaMedic, ø 19 mm, with a specific surface area (SSA) of 600 cm$^2$/cm$^3$ and a density of 0.92 g/cm$^3$), 3 L of ceramic ring media (Sera-Siporax, ø 15 mm, with a SSA of 1,000 cm$^2$/cm$^3$ and porosity < 1%) and 2 L of lava grain (AquaMedic, ø 35 mm). A pump (SunSun, 22 W, 1,000 L/h, 0.55 kg) was used to recycle the water through the filter bed, adjusted to 1,500 cm$^3$/min flow, creating a filtration speed of 2.24 cm/min. Water renewal in sump filter was less than 5% (L), caused by water evaporation and cleaning operations taking place in the aquaponic system. The circulation turnover of the aquaponic system was adjusted to eight times per hour.
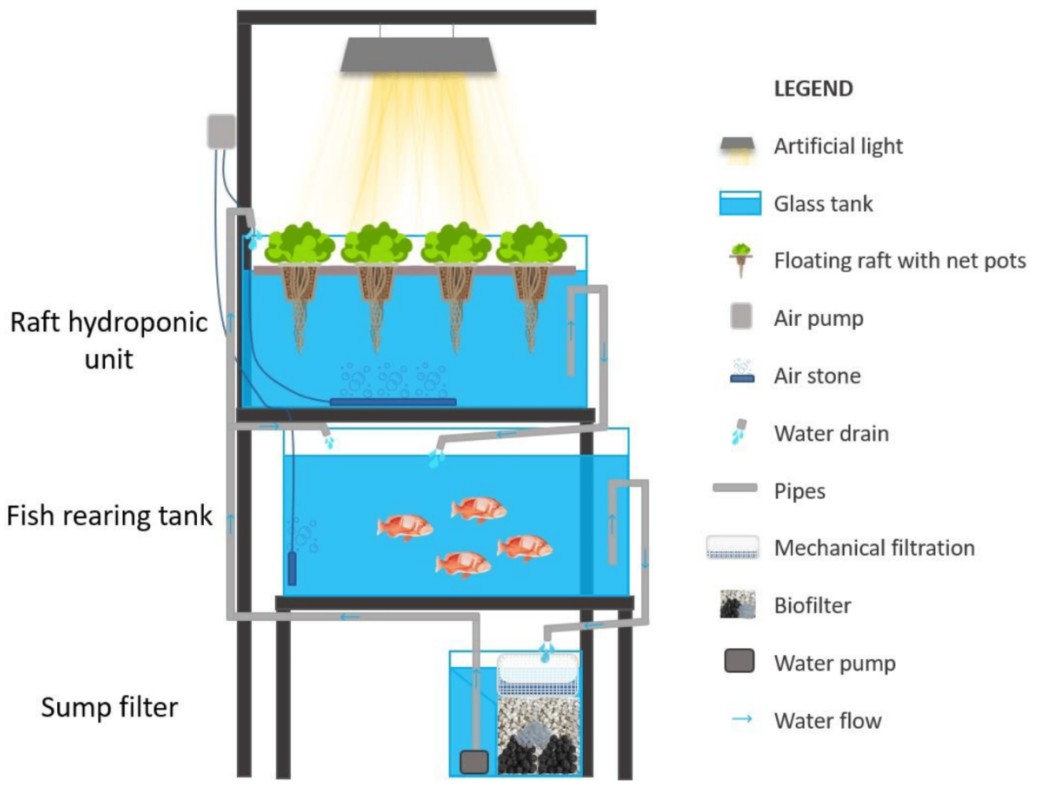

**Figure 1** Schematic side view of autonomous aquaponic system (*Tsoumalakou et al., 2023*).

In addition, 45 glasswort plants (five plants/treatment) were used in the experiment, with an average height of 3.83 ± 0.25 cm. At the end of the adaptation period, five individuals of glasswort were added to each hydroponic bed. The plants were cultivated in a floating raft system that allows the direct contact of the plant's roots with water and ensures good aeration (*Vlahos et al., 2023a*). Finally, in each aquaponic system, a 400-watt lamp (Sylvania, 400 W, high-pressure sodium) was added at a distance of 60 cm from the surface of the growth beds to provide uniform light to the plants. The photosynthetically active radiation (PAR) was maintained at 500–600 $\mu$mol m$^2$ s$^{-1}$, with a photoperiod of 14 h light/10 h dark.

## Feed intake and utilization

The feeding regime for all treatments began once the fish were stocked into the culture tanks of each aquaponic system. Both gilthead seabream and seabass were fed the experimental diet (43.7% crude protein, 15.3% crude lipid, and energy 23.2 MJ/kg) for 75 days. Food was supplied *ad libidum* (*Grigorakis & Alexis, 2005*) by hand at 09:00 h, 13:00 h, and 17:00 h. During the feeding period, fish behavior was observed and recorded daily. In each treatment, according to previous studies (*Wu et al., 2021*; *Ziegelbecker & Sefc, 2021*), a different dietary pattern was applied as follows: (i) daily feeding (0-day starvation, treatment A) (ii) feeding three days per week followed by four days of starvation (4-day

deprivation, treatment B); and (iii) feeding seven days per week followed by seven days of fasting (7-day deprivation, treatment C).

To the authors' knowledge, this is the first study investigating starvation's effect on compensatory growth and how glasswort absorbs and utilizes nutrients from fish waste. Feed was weighed daily with a precision balance to the fourth decimal place (CAS. MWP-300H), placed in tubes, and stored at 4 °C. Every 15 days, fish were anesthetized with 0.20 mg/L MSS 222 (Syndel) to re-measure the food intake and utilization. Fish tanks were cleaned daily, and uneaten food and feces were removed daily by siphoning.

Food consumption in both treatments was calculated daily by siphoning and separating the collected feces and uneaten food in the early morning and before the first meal. Feces and uneaten food were separated in a planktonic net with a 0.5−0.2 mm mesh size. The sample was thoroughly flushed with deionized water to remove salt residue, weighed, and placed in an oven at 105 °C for 24 h (*Vlahos et al., 2023b*). Food consumption was computed by the following equation described by *Vlahos et al. (2023a)*:

$$FC\,(g) = [\text{Dry weight}_{\text{food offered}} - (\text{Dry weight}_{\text{food offered}} * \text{leaching factor}/100)] - \text{Dry weight}$$
of uneaten food.

The leaching factor estimated the amount of food given during two consecutive meals and was calculated from a pre-weighed amount of 10 pellets placed in the water for 24 h and reweighed and is expressed by the formula described by *Helland, Grisdale-Helland & Nerland (1996)*:

$$\text{Leaching Factor} = [100 \times (\text{Dry weight}_{\text{food offered}} - \text{Dry weight}_{\text{food after 24h}})]/\text{Dry weight}_{\text{food offered}}.$$

The linear regression (Fig. 2) determined the correlation between the dry and wet weight of the feed given:

$$\text{Dry weight}_{\text{pellets}}\,(g) = 0,9523 \times \text{Wet weight}_{\text{pellets}} - 0,0128 \,(R^2 = 0.968,\, n = 10).$$

## Water quality criteria

The water oxygen saturation rate in both aquaponic systems was, on a mean value of 7.7 mg/L (approximately 92–93% saturation) through a cylinder air stone (10.0 × 5.1 × 5.1 cm), like all aquaponic systems. Water temperature during the experimental process in both aquaponic systems was at $20 \pm 0.5$ °C. Total ammonia (TAN), nitrite ($NO_2^-$) and nitrate ($NO_3^-$) ions, phosphate ions ($PO_4^-$), and pH were measured every 7 days using test kits (*Liddicoat, Tibhitts & Butler, 1975*). Dissolved oxygen was measured daily using a multi-parameter apparatus (HACK-LANGE, HQ 40d).

## Photosynthetic pigments content

The photosynthetic pigments of glasswort were determined spectrophotometrically after extraction with acetone. A random leaf sample of 0.6−1.0 g was selected from the plant branches, cut into pieces, and homogenized with 6–10 mL of acetone (80%) and 0.5 g of $CaCO_3$, with a mortar, pestle, and pure sand. Then, the samples were centrifuged (4,000 rpm for 20 min). The absorbance was read at 720, 663, 646, and 470 nm using a dual-beam spectrophotometer (SHIMADZU UV 1900 UV–VIS Spectrophotometer,

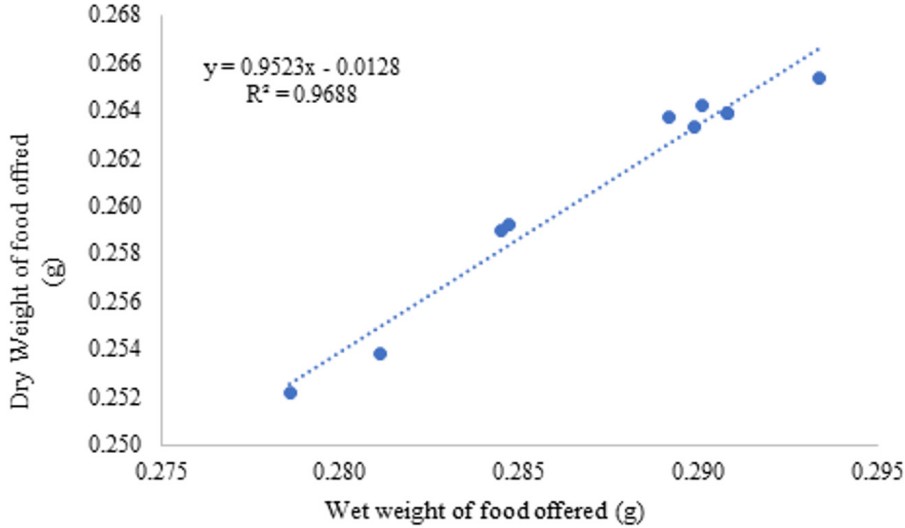

**Figure 2  Food wet weight and dry weight correlation during the 75 days of the experiment.**

Duisburg, Germany). The concentrations of chl a, chl b, and carotenoids were calculated using the equations of *Lichtenthaler & Wellburn (1983)*.

## Sampling—chemical analysis

At the beginning of the experimental process, 40 fish samples (20 individuals per species) and 40 plant samples were conducted to perform chemical analysis on the white muscle of the fish and the plant tissue, respectively. Growth assessment parameters such as total fish weight and length were taken every 15 days using a graduated ruler and an electronic balance (CAS MWP-300H). The initial and final biomass of *S. europea* was determined on the first and the last day by measuring the weight of the aerial part after drying it at 75 °C for 24 h *via* an electronic balance (CAS MWP-300H).

At the end of the experiment all fish were subjected to euthanasia. Euthanasia of animals followed the EU Directive 2010/63/EU and FELASA guidelines and performed through an overdose of Tricaine methanesulfonate (MS 222, 300+ mg/L). An approximate analysis was carried out based on the methods described by *AOAC (1995)* to determine the nutrient composition of white muscle from fish. To determine the moisture content of fish samples, the constant weight of the samples was measured after drying them in an oven at 110 °C. Crude protein content was determined by Kjeldahl analyses (nitrogen × 6.25; behr Labor-Technik, Germany), and Crude fat was determined by exhaustive Soxhlet extraction using petroleum ether (40−60 °C, BP) on a Soxtec System (C. Gerhardt GmbH & Co. KG, Königswinter, Nordrhein-Westfalen, Germany). Ash content was determined by dry ashing in porcelain crucibles in a muffle furnace at 600 °C overnight. All the above parameters are expressed as % of sample dry weight.

## Nutritional and plant growth indices

At the end of the experiment, the nutritional indices and survival rate of both cultivated species and plant growth performance were calculated from equations described by previous studies (*Endut et al., 2010*; *Stathopoulou et al., 2018*; *Karapanagiotidis et al., 2019*; *Vlahos et al., 2023a*, respectively). The following formulae were applied to the data:

Weight Gain (WG, g/fish) = FBW –IBW

Specific Growth Rate (SGR, %/days) = 100*(lnFBW –lnIBW)/days

Food consumption (g/fish) = food offered -uneaten food (g)

Feed Conversion Rate (FCR) = feed intake (g)/ wet weight gain (g)

Protein Efficiency Ratio (PER) = weight gain (gr)protein intake (g)

Fulton Condition Factor (CF) = 100* FBW (g) / $FBL^3$ (cm)

Nitrogen Retention (g*N/ABW*days) = (FBW*FBN –IBW*FBN)/(Nd*FI)

Lipid Retention (%) = 100* lipid gain(g)/lipid intake (g).

Where ABW = ((IBW + FBW)/2), IBW and FBW were the initial and final weight, IBN and FBN were the initial and final body nitrogen content, FI was the food intake and Nd was the nitrogen content in food.

The following equations of plant indices growth performance are described by *Endut et al. (2010)*.

Height Gain (LG, %) = (FH - IH) *100/IH

Weight Gain (WG, g) = FW –IW

Branch Gain (BG) = FB –IB

Yield (kg/m$^2$) = FW/Grow bed area

Where: IH and FH were the initial and final height of glasswort, IW and FW were the initial and final weight of the plant, IB and Fb were the initial and final number of glasswort branches.

## Statistical analysis

Values are presented as means ± standard error. The means from all variables obtained were analyzed for normality and variances homogeneity using Kolmogorov–Smirnov and Levene's tests, respectively. Comparison of means was performed with one-way ANOVA, which was considered statistically significant at $p < 0.05$, followed by Tukey's post-hoc test if significant differences were found, A non-parametric Kruskal-Wallis test was used when ANOVA prerequisites were not met (*Zar, 1999*). Statistical analyses were carried out using the software package IBM SPSS Statistics V27.

# RESULTS

## Water quality

The water quality variables in the different treatments are presented in Table 1. TAN, $NO_2$-, $NO_3$- and pH were measured in the outlet of the hydroponic tank, while $NO_3$- FT was measured in the fish tank. All the data followed the normal distribution ($p > 0.05$) and the variances were not significantly different from each other ($p > 0.05$). There were no significant differences between TAN, (Fig. 3) and pH in all treatments (ANOVA, $p > 0.05$), and were ranged from 0.36 ± 0.07 mg/L to 0.43 ± 0.08 mg/L for TAN, and 6.1 ± 0,08 to

**Table 1** Water chemical parameters of the trial throughout the experiment (75 days).

|  | A | B | C |
|---|---|---|---|
| TAN (mg/L) | $0.36 \pm 0.07^{a}$ | $0.43 \pm 0.08^{a}$ | $0.36 \pm 0.08^{a}$ |
| $NO_2^{-}$ (mg/L) | $0.38 \pm 0.06^{a}$ | $0.22 \pm 0.05^{ab}$ | $0.20 \pm 0.04^{b}$ |
| $NO_3^{-}$ (mg/L) | $120.6 \pm 5.82^{b}$ | $142.6 \pm 5.43^{a}$ | $138.9 \pm 3.75^{a}$ |
| $NO_3^{-}$ FT (mg/L) | $118.3 \pm 5.76^{b}$ | $136.6 \pm 5.60^{a}$ | $133.6 \pm 4.14^{a}$ |
| pH | $6.1 \pm 0.08^{a}$ | $6.2 \pm 0.05^{a}$ | $6.2 \pm 0.05^{a}$ |

Notes.

Data were expressed as mean ±SEM ($n = 20$). Means in a row followed by the same superscript are not significantly different ($p > 0.05$).

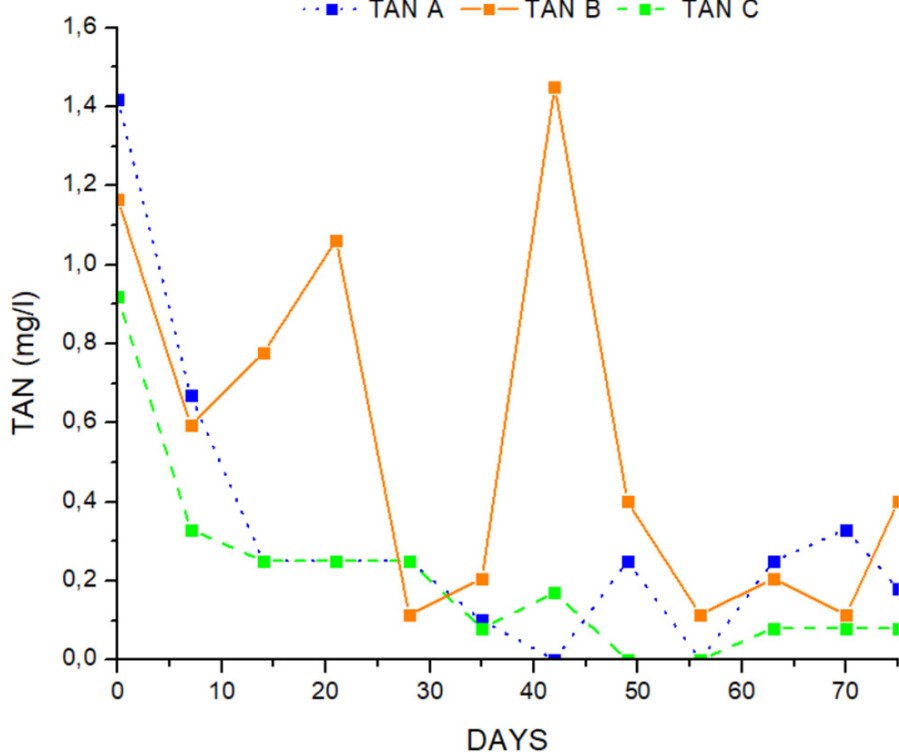

**Figure 3** Concentration of total ammonium during the experimental process.

$6.2 \pm 0.05$ for pH. Nitrites were significantly higher in treatment A than in treatments B and C and the mean value of nitrite was lower (ANOVA, $p < 0.05$) (Table 1, Fig. 4). Nitrate ions ($NO_3$) in the hydroponic tank outlet were significantly higher in treatment B and C than in treatment A (ANOVA, $p < 0.05$) (Table 1, Fig. 5) and ranged from $120.6 \pm 5.82$ mg/L for treatment A, $142.6 \pm 5.43$ mg/L for treatment B and $138.9 \pm 3.75$ for treatment C. Furthermore, $NO_3^{-}$ (mg/L) (Fig. 5) in fish tanks was significantly higher in B and C treatments than in treatment A ($136.6 \pm 5.60$, $133.6 \pm 4.14$ and $118.3 \pm 5.76$, respectively) (Table 1).

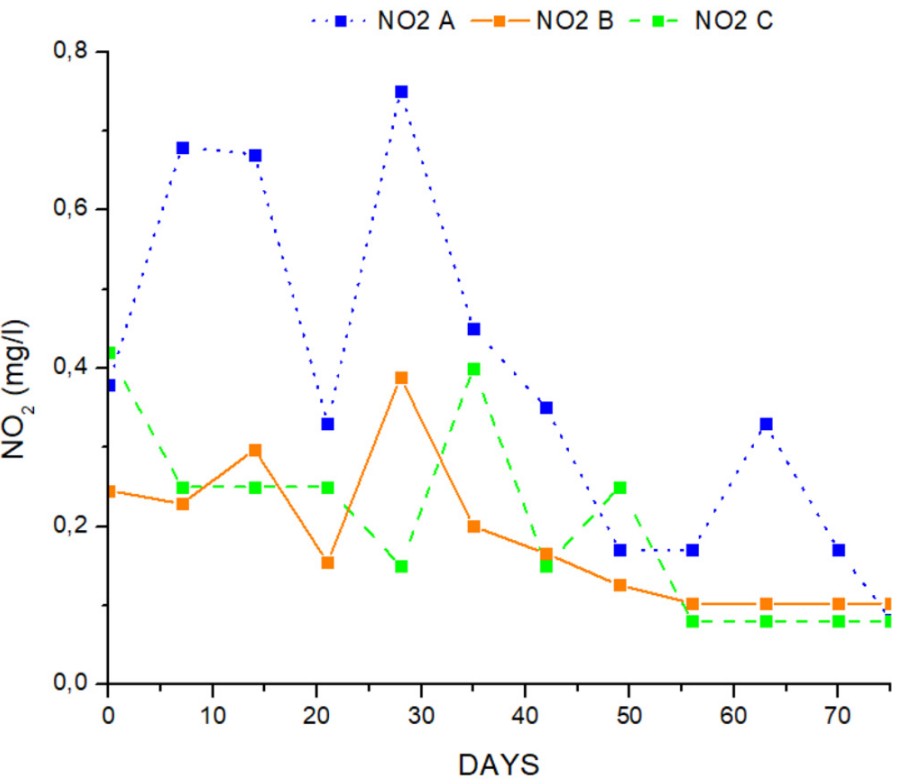

**Figure 4** Concentration of nitrite ion during the experimental process.

## Fish and plant growth performance indices

The fish growth parameters of gilthead seabream (*S. aurata*) and seabass (*D.labrax*) individuals in the integrated aquaponic system stocked in different feeding and refeeding schedules are presented in Table 2. Growth performance, feed utilization, and morphometrics data followed the normal distribution ($p > 0.05$) and the variances were not significantly different from each other ($p > 0.05$). At the beginning of the experiment, there were no significant differences in mean initial body weight (IBW) and initial body length (IBL) between treatments (ANOVA, $p > 0.05$). At the end of the experiment, the mean final body weight (FBW) of gilthead seabream and seabass was significantly higher in treatment A than in treatment C, which was lower (Table 2; ANOVA; $p < 0.05$). There were no significant differences in the final body length (FBL) between treatments (ANOVA, $p > 0.05$). The weight gain (WG) and SGR were lower at treatment B and C for both fish species (Table 2, ANOVA, $p < 0.05$). Survival rates ranged from 97% to 100% for both fish species. The Fulton condition factor (K), an indicator of lifespan, was not significantly different for both fish species per treatment (Table 2, ANOVA, $p > 0.05$). Food consumption (FC) was significantly higher in treatment A than in treatments B and C, respectively, which was lower (Table 2; ANOVA, $p < 0.05$). FCR was significantly higher in treatment C than in A and B (Table 2, ANOVA, $p < 0.05$) for both fish species (Table 2, ANOVA, $p < 0.05$). PER values were higher in treatment A for both fish species (Table 2,

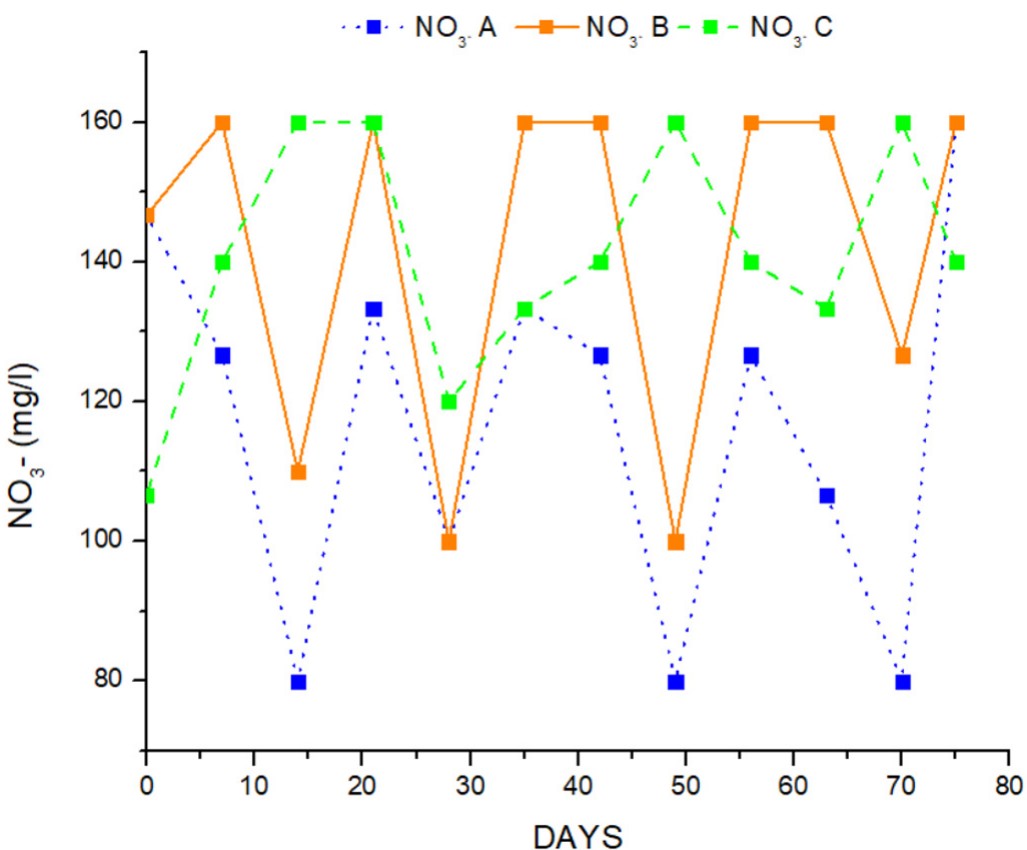

**Figure 5  Concentration of nitrate ion during the experimental process.**

ANOVA, $p < 0.05$), while nitrogen retention values were significantly higher in treatment A and lipid retention was lower for treatment C for both species (Table 2, ANOVA, $p < 0.05$), indicating lower lipid storage to fish body.

The proximate composition of the white muscle tissue of gilthead seabream and seabass individuals from the experimental trials of starvation and refeeding is shown in Table 3. All data followed the normal distribution ($p > 0.05$) and the variances were not significantly different from each other ($p > 0.05$). All groups of fish had similar white muscle protein, ash, moisture, and energy contents (Table 3, ANOVA, $p > 0.05$). Gilthead seabream individuals had significantly higher lipid content in treatment A (Table 3, ANOVA, $p < 0.05$) than in treatments B and C.

Glasswort growth performance, photosynthetic pigments content, and chemical composition data followed the normal distribution ($p > 0.05$) and the variances were not significantly different from each other ($p > 0.05$) (Table 4). Plant size at the beginning of the experiment was similar (ANOVA, $p > 0.05$) in all treatments in terms of initial height, initial biomass, and initial number of branches per plant (Table 4). Plant growth during the experiment was remarkable since the number of branches almost tripled in treatments A and B and nearly quadrupled in treatment C at the final harvest compared

**Table 2** Gilthead seabream and seabass growth performance, feed utilization, and morphometrics throughout 75 days of the experimental process in the integrated brackish aquaponic system.

| | S. aurata | | | D. labrax | | |
|---|---|---|---|---|---|---|
| | A | B | C | A | B | C |
| Survival (%) | 100 ± 0.00[a] | 100 ± 0.00[a] | 97 ± 3.34[a] | 100 ± 0.00[a] | 100 ± 0.00[a] | 97 ± 3.30[a] |
| IBW (g) | 6.35 ± 0.15[a] | 6.32 ± 0.09[a] | 6.32 ± 0.13[a] | 5.84 ± 0.09[a] | 5.77 ± 0.11[a] | 5.84 ± 0.12[a] |
| IBL (cm) | 5.73 ± 0.13[a] | 5.86 ± 0.11[a] | 5.77 ± 0.13[a] | 6.35 ± 0.08[a] | 6.34 ± 0.12[a] | 6.37 ± 0.14[a] |
| FBW (g) | 27.09 ± 0.73[a] | 22.38 ± 0.51[b] | 20.64 ± 0.77[b] | 26.05 ± 0.75[a] | 22.5 ± 0.57[b] | 20.55 ± 0.62[b] |
| FBL (cm) | 11.69 ± 0.56[a] | 11.62 ± 0.65[a] | 10.98 ± 0.49[a] | 12.11 ± 0.40[a] | 11.08 ± 0.24[a] | 11.96 ± 0.36[a] |
| WG (g) | 20.75 ± 0.78[a] | 16.06 ± 0.49[b] | 14.32 ± 0.81[b] | 19.7 ± 0.77[a] | 16.16 ± 0.61[b] | 14.18 ± 0.69[b] |
| SGR (% /day) | 2.07 ± 0.06[a] | 1.80 ± 0.04[a] | 1.67 ± 0.07[b] | 2.00 ± 0.04[a] | 1.80 ± 0.05[a] | 1.66 ± 0.06[b] |
| FC (g) | 2.33 ± 0.05[a] | 1.38 ± 0.08[c] | 1.95 ± 0.05[b] | 2.34 ± 0.05[a] | 1.37 ± 0.08[c] | 1.98 ± 0.05[b] |
| FCR | 1.31 ± 0.08[b] | 1.11 ± 0.09[b] | 2.03 ± 1.16[a] | 1.42 ± 0.09[b] | 1.09 ± 0.09[b] | 1.97 ± 0.13[a] |
| PER (%) | 0.94 ± 0.04[a] | 0.69 ± 0.02[a] | 0.61 ± 0.03[b] | 0.90 ± 0.03[a] | 0.69 ± 0.02[a] | 0.61 ± 0.03[b] |
| Lipid retention (%) | 13.77 ± 0.00[a] | 9.19 ± 0.00[b] | 7.65 ± 0.00[c] | 7.44 ± 0.48[a] | 5.36 ± 0.01[b] | 4.39 ± 0.03[c] |
| Nitrogen retention (gr N/ kg ABW/ day) | 1.21 ± 0.02[a] | 1.13 ± 0.02[a] | 1.07 ± 0.03[b] | 1.19 ± 0.02[a] | 1.12 ± 0.02[ab] | 1.07 ± 0.03[b] |
| K (g/cm³ ) | 2.67 ± 0.44[a] | 2.47 ± 0.42[a] | 1.97 ± 0.21[a] | 1.67 ± 0.14[a] | 1.8 ± 0.12[a] | 1.41 ± 0.14[a] |

Notes.
Data were expressed as mean ± SEM ($n = 180$). Means in a row followed by the same superscript are not significantly different ($p > 0.05$).
Abbreviations: CF, condition factor; FBW, final body weight; FC, food consumption; FCR, feed conversion ratio; IBW, initial body weight; PER, protein efficiency ratio; SGR, specific growth rate; WG, weight gain; IBL, initial body length; FBL, final body length.

**Table 3** Approximate analysis of the muscle of gilthead seabream and seabass at the end of cultivation (75 days) in the polyculture aquaponic systems.

| | S. aurata | | | D. labrax | | |
|---|---|---|---|---|---|---|
| | A | B | C | A | B | C |
| Protein % | 68.37 ± 0.29[a] | 69.52 ± 0.24[a] | 70.03 ± 0.68[a] | 68.12 ± 0.41[a] | 69.26 ± 0.37[a] | 70.18 ± 0.37[a] |
| Lipid % | 11.08 ± 0.16[a] | 10.41 ± 0.05[b] | 9.41 ± 0.25[c] | 10.56 ± 0.35[a] | 10.37 ± 0.03[a] | 9.35 ± 0.25[a] |
| Ash % | 8.76 ± 0.37[a] | 9.41 ± 0.56[a] | 7.76 ± 0.62[a] | 9.07 ± 0.01[a] | 9.56 ± 0.36[a] | 9.79 ± 0.68[a] |
| Moisture (%) | 26.29 ± 0.39[a] | 26,32 ± 0.41[a] | 24.74 ± 0.76[a] | 26.29 ± 0.52[a] | 25.09 ± 0.67[a] | 24.23 ± 0.52[a] |
| Energy (MJ/kg) | 22.41 ± 0.01[a] | 22.23 ± 0.09[a] | 22.32 ± 0.05[a] | 22.23 ± 0.09[a] | 22.18 ± 0.05[a] | 21.98 ± 0.15[a] |

Notes.
Data were expressed as mean ± SEM ($n = 180$). Means in a row followed by the same superscript are not significantly different (ANOVA, $p > 0.05$).

with the initial values, without significant differences among treatments (Table 4, ANOVA, $p > 0.05$). Glasswort height growth (HG) was significantly higher in treatment C than in treatments A and B (Table 4, ANOVA, $p < 0.05$). Plant weight gain (WG) was lower in treatment A, while branch gain (BG) was higher in treatment C (Table 4, ANOVA, $p < 0.05$). Throughout the entire 75-day experimental period, the yield of glasswort was noticeably lower in treatment A than in treatment B and higher in treatment C (Table 4, ANOVA, $p < 0.05$).

The effects of the nutritional treatments on the concentration of the plant's photosynthetic pigments are also presented in Table 4. By the end of the experiment, chlorophyll a was found to be significantly higher in treatment A, while chlorophyll b was similar for all treatments (Table 4, ANOVA, $p > 0.05$). Carotenoids at the end of

**Table 4 Glasswort growth performance, photosynthetic pigments content, and chemical composition at the end of the cultivation period (75 days).**

| | A | B | C |
|---|---|---|---|
| Initial height (cm) | 4.00 ± 0.26[a] | 3.77 ± 0.30[a] | 3.73 ± 0.18[a] |
| Initial biomass (g) | 1.04 ± 0.12[a] | 0.94 ± 0.09[a] | 0.76 ± 0.08[a] |
| Final height (cm) | 34.04 ± 0.59[b] | 31.45 ± 1.11[b] | 37.49 ± 0.86[a] |
| Final biomass (g) | 33.71 ± 1.85[c] | 64.50 ± 5.73[b] | 87.57 ± 8.45[a] |
| The initial number of branches | 1.73 ± 0.15[a] | 1.80 ± 0.20[a] | 1.40 ± 0.13[a] |
| The final number of branches | 5.27 ± 0.25[a] | 5.73 ± 0.23[a] | 6.00 ± 0.24[a] |
| HG (cm) | 30.04 ± 0.64[b] | 27.69 ± 1.02[b] | 33.75 ± 0.89[a] |
| WG (g) | 32.67 ± 1.85[c] | 63.57 ± 5.71[b] | 86.81 ± 8.45[a] |
| BG | 3.53 ± 0.22[b] | 3.93 ± 0.23[ab] | 4.6 ± 0.21[a] |
| Yield (kg/m$^2$) | 1.87 ± 0.10[c] | 3.58 ± 0.32[b] | 4.87 ± 0.47[a] |
| Photosynthetic pigments | | | |
| Chl a (mg/g) | 2.03 ± 0.17[a] | 1.65 ± 0.85[ab] | 1.35 ± 0.11[b] |
| Chl b (mg/g) | 0.49 ± 0.37[a] | 0.38 ± 0.03[a] | 0.42 ± 0.03[a] |
| Carotenoids (mg/g) | 0.55 ± 0.03[a] | 0.36 ± 0.4[b] | 0.59 ± 0.04[a] |
| Chemical composition | | | |
| Protein (%) | 14.62 ± 0.27[a] | 13.87 ± 0.85[a] | 14.34 ± 0.2[a] |
| Ash (%) | 14.47 ± 0.05[a] | 13.93 ± 0.08[b] | 13.84 ± 0.11[c] |
| Moisture (%) | 79.31 ± 4.39[b] | 85.59 ± 5.12[a] | 86.75 ± 5.23[a] |

**Notes.**

Data were expressed as mean ± SEM ($n = 54$). Means in a row followed by the same superscript are not significantly different ($p > 0.05$).

Abbreviation: HG, Height gain; WG, weight gain; BG, branches gain; Chl a, Chlorophyll A; Chl b, Chlorophyll b.

the experiment were found to be significantly higher in treatments A (0.55 ± 0.03 mg/g) and C (0.59 ± 0.04 mg/g) than in treatment B (0.36 ± 0.4 mg/g) (ANOVA, $p < 0.05$). Regarding plant tissue protein, the results showed that there were no significant differences between treatments at the end of the experiment (Table 4, ANOVA, $p > 0.05$), in contrast to moisture content, which was significantly lower in treatment A (Table 4, ANOVA, $p < 0.05$), while ash was significantly higher in this treatment.

# DISCUSSION

An innovative approach to aquaponics systems is presented in the present study, which compares an innovative coupled polycultures aquaponics system on a small scale by implementing technical practices based on compensatory techniques as a factor affecting growth performance and food utilization throughout the culture period. Most studies on compensatory growth (*Jobling, 2010*) in fish have examined responses to changes in feed availability; few studies (*Allsopp, De Lange & Veldtman, 2008*; *Castagna et al., 2022*) have addressed the effects of feed quality, temperature, or fish density on fish growth performance. Compensatory growth is considered a promising tool to increase aquaculture and aquaponics production. It has been widely evaluated in farmed fish due to its faster growth rate and better feed utilization. The fish can be kept under restricted feeding

conditions for longer (*Ali, Nicieza & Wootton, 2003*; *Krogdahl & Bakke-McKellep, 2005*; *Jobling, 2010*).

## Water quality

Water quality indices such as TAN, and $NO_2^-$ were maintained at low levels throughout the experimental procedure. No statistically significant differences were detected in either treatment (ANOVA, $p > 0.05$). The mean values of $NO_3^-$ at the inlet checkpoint of the hydroponic subsystem were higher than at the outlet checkpoint, indicating that glasswort was taking nutrients through water. However, water quality parameters were within the optimal range for aquaculture (*Boyd & Tucker, 2014*) and aquaponics systems (*Somerville et al., 2014*). The present study showed that compensatory growth effectively reduced ammonia concentrations and increased nitrate levels (*Das, Tanmoy & Mandal, 2012*). Plants use the increased nitrate ion levels for nutrition and development (*Somerville et al., 2014*).

Ammonia is a suitable source of nutrients and was taken up by plants when nitrate levels were low. In contrast, plants absorb much more nitrate when ammonia levels increase (*Xu, Tsai & Tsai, 1992*). According to *Buzby & Lin (2014)*, the retention efficiency of ammonia ($NH_4^+/NH_3$) was higher than that of nitrate ions in a freshwater aquaponic system where lettuce was grown. The results of the present study showed that the ammonia concentration was maintained low throughout the experimental period, indicating that the same amount of ammonia oxidized to nitrite and nitrate ions was achieved in all treatments and was suitable for the uptake and growth of glasswort plants. *Quinta (2013)* reported levels of $116.71 \pm 17.57$ mg/L for nitrate ions and $0.2 \pm 0.02$ mg/L for ammonia.

The findings of the present study showed that nitrate ions ($NO_3-$) at the outlet checkpoint from the growing beds varied from $118.33 \pm 5.76$ mg/L in treatment A to $136.66 \pm 5.61$ mg/L in treatment B and $133.66 \pm 4.15$ ml/L in treatment C, indicating efficient functioning of the coupled aquaponic system and oxidation capacity of the filter (*Spotte, 1992*; *Somerville et al., 2014*; *Vlahos et al., 2023a*). Previous studies reported lower levels of nitrate ions ($93.94 \pm 7.81$ mg/L to $119.81 \pm 7.36$ mg/L) (*Vlahos et al., 2023b*) than the nitrate levels found in the present study in a brackish water aquaponic system in which seabass and rock samphire (*Crithmum maritimum*) were cultured under three different salinities (8 ppt, 14 ppt, 20 ppt). In addition, *Vlahos et al. (2023a)* reported much lower nitrate concentrations ranging from 76.4 mg/L to 77.2 mg/L in a brackish aquaponic system with recirculation and co-culture of seabass and rock samphire at salinities of 8 ppt and 14 ppt, respectively.

In our study, pH values were similar among treatments. Plants require a pH between 5.5 and 6.5 to improve nutrient uptake. The best pH range for bacteria is 7.0−8.0, while the recommended pH range for aquaculture is 6.5−8.5 (*Yildiz et al., 2017*). Thus, the optimal pH range for the aquaponic system appears to be 6.5−7.0. pH values above 7.0 may result in decreased solubility of phosphorus and nutrients. The uptake of certain nutrients by plants is limited in the aquaponic environment (*Tyson et al., 2004*).

## Fish growth performance

The results of fish growth performance showed a partial positive effect of starvation–refeeding on compensatory growth. They did not indicate detrimental compensatory growth in starved gilthead seabream or seabass (treatments B and C) relative to daily feeding fish (treatment A). The deprived gilthead seabream and seabass (treatment C) showed significantly lower growth performance. Fish refeeding is a factor that should be considered in compensatory growth. In cases of total compensation, the deprived fishes eventually reach the same size at the same age as their continuously fed counterparts (*Ali, Nicieza & Wootton, 2003*; *Krogdahl & Bakke-McKellep, 2005*; *Jobling, 2010*).

Our findings are consistent with those of *Jobling (2010)* for Atlantic cod *Gadus morhua*, where weight did not show a full compensatory effect after three weeks of refeeding. *Tunçelli & Pirhonen (2021)* also reported statistically insignificant compensation in their growth performance when rainbow trout was starved over the weekends. In the present study, both fish species grown in a polyculture aquaculture system and fed daily (treatment A) showed the highest weight gain and specific growth rate (ANOVA, $p < 0.05$).

Fish survival at the end of the experiment varied between 97% and 100% and was higher than the survival reported by *Hasanpour et al. (2021)* for yellowfin seabream *Acanthopagrus latus* (96.7–100%) and Sobaity seabream *Sparidentex hasta* (83–96%). Fulton's condition factor (K) is an indicator of lifespan and showed no significant differences between treatments for either species in the aquaponic systems (ANOVA, $p > 0.05$), suggesting that short-term food deprivation does not significantly affect fish welfare. Fulton's condition factor value at the end of the rearing period (75 days) was compared with the condition factor value found in other studies. Our value was higher than the K found for Atlantic cod *Gadus morhua* (K: $1.31 \pm 0.11$ g/cm$^3$, $1.27 \pm 0.11$ g/cm$^3$, and $1.22 \pm 0.11$ g/cm$^3$) when one-, two-, and three-week starvation was followed by a one-week refeeding schedule (*Jobling, 2010*). *Hasanpour et al. (2021)* reported a much lower K (1.1 to 1.4 g/cm$^3$) for *Acanthopagrus latus* restricted to two weeks of refeeding.

This indicates that the physical condition of fish in this polyculture system appears to be significantly affected by the controlled rearing environment, promoting the growth of gilthead seabream more than that of seabass (ANOVA, $p > 0.05$). In addition, red pagrus *Pagrus pagrus* exhibited a lower K ($2.53 - 2.55$ gr/cm$^3$) after 14 days of starvation and 7 or 15 days of refeeding than in the present experiment (*Caruso et al., 2012*). In turn, *Pérez-Jiménez et al. (2012)* reported that common dentex (*Dentex dentex*), when exposed to a starvation–refeeding regime, showed a K of $1.61 - 1.74$ g/cm$^3$, which was similar to the K of *Dicentrarchus labrax* found in the current experiment. The K is a way to verify the level of energy reserves as well as the health status of fish, and changes in its value may indicate changes in the nutritional status of the fish (*Goede, 1990*).

Starvation has been reported to cause oxidative stress in fish, which reduces the storage of nutrients and antioxidants such as sulfur-containing amino acids (*e.g.*, methionine, cysteine, and taurine) and glutathione in fish, negatively affecting the activity of glutathione-dependent enzymes. Furthermore, the results showed that gilthead seabream and seabass, when cultured in a brackish water aquaponic system and fed daily (0 days of starvation),

showed a significantly increased mean final weight, in contrast to the values found in treatments B (4 days of starvation) and C (7 days of starvation).

The results of the present study are consistent with those of *Eroldoğan, Taşbozan & Tabakoğlu (2008)*, in which gilthead seabream with an average initial weight of $5.85 \pm 054$ g showed an increased mean final weight ($27.08 \pm 0.8$ g) after 8 days of feeding followed by 2 days of starvation. Gilthead seabreams were reported to show a fourfold increase in the mean final weight ($23.49 \pm 1.51$ g and $22.90 \pm 1.16$ g) when starved for 5 and 20 days, respectively (*Eroldoğan, Taşbozan & Tabakoğlu, 2008*). Other studies reported that *Salmo salar, Aristichthys nobilis*, and *Gasterosteus aculeatus* reached the same mean weight as daily fed species after a starvation and refeeding program (*Maclean & Metcalfe, 2001*; *Xie, 2001*; *Zhu et al., 2001*). Fish species such as gilthead seabream (*Eroldoğan, Taşbozan & Tabakoğlu, 2008*), sturgeon *Acipenser dabryanus* (*Wu et al., 2021*), and Atlantic salmon (*Hvas et al., 2022*) grow faster after a period of starvation and refeeding. Moreover, our findings showed significantly lower specific growth rate (SGR) and weight gain (WG) under the 4-day starvation (treatment B) and 7-day starvation (treatment C) conditions for seabream and seabass compared to treatment A (0 days of starvation, daily feeding) in the brackish polyculture aquaponic system. *Nikki et al. (2004)* reported that rainbow trout (*Oncorhynchus mykiss*) showed similar SGR under two (SGR: 2%/d), four (SGR: 2.2%/d), eight (SGR: 1.7%/d), and 14 (SGR: 1.7%/d) days of starvation to fish in the continuous feeding treatment (SGR: 2.1%/d).

Factors such as feed intake, fish size, and water temperature influence the digestibility of nutrients in the feed and, thus, the specific growth rate. Compensatory growth experiments have reported mechanisms such as hyperphagia (*Ali, Nicieza & Wootton, 2003*), metabolic rate optimization (*Alvarez & Nicieza, 2005*), protein biosynthesis (*Quinton & Blake, 1990*), endocrine system adaptability during restricted feeding (*Davis & Gaylord, 2011*), basal metabolic rate reduction, and FCR improvement (*Mozanzadeh et al., 2020*) to be at play.

Food consumption showed significant differences, with the highest value observed in gilthead seabream and seabass when a 7-day starvation and a 7-day refeeding period were implemented (ANOVA, $p > 0.05$). In compensatory growth experiments, *Eroldoğan, Kumlu & Sezer (2006b)* reported higher food consumption in gilthead seabream ($34.9 \pm 0.63$ g) when exposed to starvation for 2, 4, and 7 days. Restriction of food consumption leads to biochemical changes and a reduction in the metabolic rate of fish, resulting in temporary weight loss, but the fish subsequently acclimate and adapt to this rate and gain weight (*Castagna et al., 2022*). *Hepher et al. (1983)* reported that red tilapia, *Oreochromis spp.* (Cichlidae), readjusted their metabolic rate and gained weight after acclimation to restricted food consumption. Food consumption is related to the rate of gut absorption and excretion.

The feed conversion ratio (FCR) was significantly lower in treatments A and B than in treatment C (ANOVA, $p < 0.05$). *Türkmen et al. (2012)* reported a lower FCR ($1.0 \pm 0.04 - 1.2 \pm 0.75$) for seabass. *Yılmaz & Eroldoğan (2011)* reported that juvenile gilthead seabream subjected to two starvation periods with $1 + 3$ refeeding (1 day of starvation, 3 days of refeeding) and $1 + 5$ refeeding (1 day of starvation, 5 days of refeeding) showed hyperphagia after each starvation period. Compensatory growth feeding
is associated with hyperphagia in fish. A decrease in food passage through the digestive tract leads to hyperphagia. Hyperphagia allows an animal to achieve the same or even higher total feed consumption than an individual with continuous access to food and thus achieve the same size (*Das, Tanmoy & Mandal, 2012*).

Furthermore, food consumption (FC) and specific growth rate (SGR) are allometrically related to the weight of fish (*Wootton, 1998*; *Jobling, 2010*). In the present study, the final body weight (FBW) of gilthead seabream and seabass were significantly lower in treatments B and C, respectively. These fish exhibited a higher FC and SGR than fish fed daily (treatment A) due to the allometric interrelation between food consumption and fish weight.

The protein efficiency ratio (PER) showed significant differences among treatments (ANOVA, $p < 0.05$) in both fish species, suggesting that both species were affected by the feeding schedule applied throughout the experiment. Similar results were obtained by *Ali & Jauncey (2004)*, where PER values differed between the control and starvation treatments. In contrast, *Cho et al. (2006)* reported that PER values did not vary between the control and 7-day starvation treatments. However, in the feeding regime schedule that was applied for more days of starvation (2, 3, and 4 weeks), the PER values decreased as the days of starvation increased.

Lipid retention (LR) showed significant differences in both fish species among treatments (ANOVA, $p < 0.05$), with the highest value found in treatment A and the lowest in treatment C. Our findings are comparable to those reported by *Karapanagiotidis et al. (2021)*, although *Diplodus puntazzo* showed higher LR values. Notably, *Karapanagiotidis et al. (2021)* reported that in fish fed the diet with the lowest protein content (30%), the LR value ranged from $34.25 \pm 1.57\%$ to $77.64 \pm 3.22\%$, while the LR values found in the fish fed the diet containing 45% crude protein were similar to those found for the diet used in the present study (43.7% crude protein). During food deprivation, energy consumption decreases due to a reduction in the fish's locomotor activity. In the refeeding period, reduced activity contributes to compensatory growth in fish, with the available energy used for growth. When fish species such as seabass *Dicentrarchus labrax* (*Stirling, 1976*), rainbow trout *Oncorhynchus mykiss* (*Jezierska, Hazel & Gerking, 1982*), and tilapia *Oreochromis niloticus* (*Satoh, Takeuchi & Watanabe, 1984*) are under starvation, they breakdown lipids faster than other nutrients. Changes in lipid levels occur before and during compensatory growth. *Miglavs & Jobling (1989)* reported a decreased liver/spleen-to-body mass ratio in Arctic char *Salvelinus alpinus* when starvation was applied for 8, 12, and 16 weeks. Arctic char also showed an inverse relationship between food intake (% body weight/day) and carcass lipids (% wet weight) and visceral lipids (% wet weight) (*Jobling, 2010*). Changes in lipid levels occur before and during compensatory growth through decreases in liver and viscera as proportions of body mass and take place in fish as a response to food restriction, which is dependent on the number of fasting days (*Miglavs & Jobling, 1989*).

Nitrogen retention (NR) was significantly lower in treatment C than in treatment A (ANOVA, $p < 0.05$) for both fish species. In juveniles *Diplodus puntazzo*, NR values ranged from 0.12 to 0.32 g/kg/ABW/day and were lower than the results of the present study (*Coutinho et al., 2012*). According to *Bastrop, Spangenberg & Jürss (1991)*, during

food deprivation, *Cyprinus carpio* shows a depression of 43% in the RNA/DNA quotient of white muscle cells after starvation for 28 days at a water temperature of 22 °C, which indicates a reduced level of metabolism in the muscle tissue and reduced overall fish growth. To reduce the total protein concentration, protein synthesis in the fish was decreased compared to the treatments in which fish were fed daily. Protein and ash levels showed no significant differences in either species (ANOVA, $p > 0.05$), in contrast to total lipids, which showed significant differences in gilthead seabream (ANOVA, $p < 0.05$). The biochemical composition of the fish body was not affected by the dietary schedule followed to achieve compensatory growth (*Gaylord & Gatlin, 2000*; *Zhu et al., 2004*; *Turano, Borski & Daniels, 2007*). However, the total lipid content had a tendency to be lower in treatment C than in treatments A and B for both fish species. Lipids are essential for basal metabolism and survival during prolonged starvation (*Adaklı & Taşbozan, 2015*). Similarly, studies have reported that the decrease in crude lipids is a consequence of starvation (*Oh, Noh & Cho, 2007*; *Wang et al., 2009*; *Peres, Santos & Oliva-Teles, 2011*).

## Plant growth performance

Glasswort is a halophyte that thrives in saline environments. Halophytes employ various processes and metabolic modifications that allow them to grow in a wide range of soil salinities, from the ordinarily low levels found in agricultural soils to the extremely high levels found in coastal ecosystems (*Cabot et al., 2014*). The responses to salinity are species-specific, and the salinity level for optimal growth is determined by genotype and environmental conditions. Many halophytes grow better when exposed to mild or moderate salinity due to established physiological adaptations (*Winicov & Bastola, 1997*; *Kumar et al., 2019*). Glasswort belongs to this group since it shows maximum growth between 12 and 25 ppt, and salinity levels below or above this range (*i.e.,* 0.55 and 72 ppt) decrease its growth performance (*Cárdenas-Pérez et al., 2022*). Likewise, *Rozema & Schat (2013)* reported that 10–22 ppt favoured the growth of glasswort. Therefore, the salinity levels used in the present study fall within the limits of optimal growth of glasswort. The fish-feeding treatments significantly impacted several plant growth parameters.

Notably, our results showed a positive effect of fish starvation–refeeding in glasswort growth performance. Glasswort cultivated under treatment C showed higher biomass accumulation and increased height, with higher branch gain than the other two treatments. Overall, the plant yield under treatment C exhibited a 3.6-fold and 1.4-fold increase compared to treatments A and B, respectively. The increased $NO_3$ in the recirculating water may partly account for this yield increase. In contrast to our results, *Tuncelli & Memiş (2024)* reported that that increased feeding improves plant growth in aquaponic systems. Specifically, in an aquaponic system with rainbow trout and lettuce, when fish exposed to current-assisted swimming and fed daily until satiation, plant weight exhibited a significantly increment. *De Souza et al. (2018)* cultivated the congeneric *Salicornia neei* in a greenhouse under 11-ppt salinity. They found a similar number of branches but slightly taller plants than in our study. Regarding productivity, the yield observed in treatment C ($4.87 \pm 0.47$ kg/m$^2$) significantly outweighed the yields reported by *Pinheiro et al. (2020)*

for the congeneric *Sarcocornia ambigua* (glasswort) co-cultivated with pacific white shrimp *Litopenaeus vannamei* at 16-ppt salinity.

The growth superiority of plants under treatment C cannot be ascribed to their leaf protein content, which remained unaffected by the fish-feeding treatments. However, the protein levels determined in the present study indicated that *Salicornia europaea* is a valuable source of proteins. Comparable contents of crude protein were reported in *Salicornia europaea* by *Castagna et al. (2022)* (12.8–13.8%), in *Atriplex lentiformis* by *Díaz, Benes & Grattan (2013)*, and in *Arthrocnemum macrostachyum* by *Barreira et al. (2017)*. Moreover, a significantly lower ash content was found in treatments B and C than in the control. Overall, the ash content of the present study was lower than the values reported by *Castagna et al. (2022)* for the same species (31.2%). The high ash content is a characteristic of halophytes and indicates the higher amount of minerals the plants deposit in their tissues.

Concerning photosynthetic pigments, the results showed significantly lower chlorophyll a and carotenoid concentrations in treatment C. Chlorophyll b and carotenoid concentrations were similar among treatments. The observed reductions in chlorophyll content agreed with those reported by *Cárdenas-Pérez et al. (2022)* at salinity levels above 400 mM NaCl. According to *Winicov & Bastola (1997)*, chlorophyll and carotenoid contents are lower in cultivated than in field-collected halophytes.

## CONCLUSIONS

To the best of our knowledge, the present study is the first to investigate the effect of compensatory growth on the growth performance and survival rate of gilthead seabream, seabass, and glasswort in a brackish polyculture aquaponic system. The results showed that cultivation of glasswort in the polyculture aquaponic system with brackish water is advantageous, as it efficiently absorbs the nutrients produced by the co-cultured fish (gilthead seabream and seabass) in the system.

The effect of food deprivation (long-term period: 7 days and short-term period: 4 days) was significantly beneficial for the growth performance of glasswort compared to the control treatment. Glasswort has significant potential for exploitation due to its increased commercial interest. It can be used as a biofuel and as a crop in areas with high salinity and nutrient-poor substrates. However, when seven days of food deprivation were applied, gilthead seabream and seabass showed significantly lower growth. However, the growth performance of glasswort was higher, increasing its biomass by 135%, indicating that it utilized the available nutrients in the aquaculture system more efficiently.

The results suggested that the feeding schedule involving starvation–refeeding cycles could be a promising feed management option for the co-cultivation of different species with plants in a brackish aquaponic system. Further research on physiological and metabolic responses is needed to thoroughly understand the role of each species used in aquaponic systems. Testing different starvation-refeeding cycles on different fish species and plants will be helpful.

## ACKNOWLEDGEMENTS

The authors wish to express their gratitude to PHILOSOFISH SA and ZOONOMI SA for generously supplying the gilthead seabream (*S. aurata*), sea bass (*D.labrax*) and fish feed used in the experiment, respectively.

### Funding

The authors received no funding for this work.

### Competing Interests

Efi Levizou is an Academic Editor for PeerJ.

### Author Contributions

- Ioannis Mitsopoulos conceived and designed the experiments, performed the experiments, analyzed the data, prepared figures and/or tables, authored or reviewed drafts of the article, and approved the final draft.
- Iliana Gesthimani Kontou performed the experiments, analyzed the data, authored or reviewed drafts of the article, and approved the final draft.
- Konstantinos Babouklis performed the experiments, authored or reviewed drafts of the article, and approved the final draft.
- Nikolaos Vlahos conceived and designed the experiments, performed the experiments, analyzed the data, prepared figures and/or tables, authored or reviewed drafts of the article, and approved the final draft.
- Panagiotis Berillis conceived and designed the experiments, analyzed the data, prepared figures and/or tables, authored or reviewed drafts of the article, and approved the final draft.
- Efi Levizou analyzed the data, prepared figures and/or tables, authored or reviewed drafts of the article, and approved the final draft.
- Eleni Mente conceived and designed the experiments, analyzed the data, prepared figures and/or tables, authored or reviewed drafts of the article, and approved the final draft.

### Animal Ethics

The following information was supplied relating to ethical approvals (i.e., approving body and any reference numbers):

Ethics Committee of the Region of Thessaly, Veterinary Directorate, Department of Animal Protec-tion-Medicines-Veterinary applications (n. 112841/23-03-2022).

### Data Availability

Water quality data, Fish and Plant Growth Performance data are available in the Supplementary Files.

## Supplemental Information

Supplemental information for this article can be found online at http://dx.doi.org/10.7717/peerj.17814#supplemental-information.

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
