# Peer review of "Starvation and re-feeding of Gilthead seabream (Sparus aurata) and European seabass (Dicentrarchus labrax) co-cultured with glasswort (Salicornia europaea) in a polyculture aquaponic system"

_PeerJ, doi:10.7717/peerj.17814_

## Round 0.1 · original submission · Minor Revisions

Dear Dr. Berillis

The reviewers have commented on your manuscript. Based on the comments and suggestions of the expert reviewers, a minor revision is needed for your article.

I would like to request that you check and correct the manuscript based on the reports.

Sincerely yours

Reviewer 1 ·

Basic reporting

General Comment:
The manuscript is well-structured and detailed, covering a range of pertinent aspects of the study. It explores an innovative approach to aquaponics that integrates multiple species, which is a strength. However, there are several critical areas that need improvement to enhance the manuscript's clarity, impact, and its contribution to the field.
Specific Comments:
The current introduction effectively outlines the uniqueness and importance of the topic, yet it falls short in explicitly defining the study's objectives and hypotheses. To fortify the rationale and enhance the contextual relevance of this research within the field, it is essential to clearly state the research gaps that this study aims to address. Following the approach of Tunçelli and Pirhonen (2021), the introduction should be revised to include a distinct statement of the study’s hypothesis, thereby delineating the specific issues the research seeks to explore and the anticipated contributions to the existing body of knowledge.
The description of the methods is thorough, providing clarity on the experimental setup and protocols used. It would be beneficial for the authors to include any limitations of their study design that could influence the interpretation of the results.
The statistical methods are adequately described. However, it would be helpful if the authors could discuss the assumptions of the statistical tests used and confirm that these assumptions were met in their data.
The section on Results and Discussion effectively presents and analyzes the findings. However, to deepen the analysis, it would be beneficial to extensively compare these results with prior research, specifically addressing any variations and offering potential explanations for these discrepancies. Additionally, incorporating references to recent studies that present contrasting results, such as those suggesting that increased feeding improves plant growth in aquaponic systems (e.g., Tunçelli & Memiş, 2024), would enhance the richness of the discussion. This approach not only broadens the perspective but also enriches the dialogue by integrating diverse viewpoints and current debates within the field.
The manuscript seems well-cited, but the authors should ensure that all references are up-to-date and relevant, especially focusing on recent developments in the field.
There are minor grammatical and punctuation errors throughout the text that need to be corrected through thorough proofreading.
The conclusion succinctly summarizes the findings and their implications. It could be strengthened by suggesting specific future research directions based on the results obtained.
The manuscript inconsistently uses two different citation styles, which could lead to confusion. For instance, in the statement "Most studies on compensatory growth (Jobling 2010) in fish have examined responses to changes in feed availability; few studies [4,16] have addressed the effects of feed quality, temperature, or fish density on fish growth performance. Compensatory growth is considered a promising tool to increase aquaculture and aquaponics production. It has been widely evaluated in farmed fish due to its faster growth rate and better feed utilization. The fish can be kept under restricted feeding conditions for longer (Ali et al. 2003; Kroghahl and Bakke-McKellep 2005; Jobling 2010)," the use of both APA style and numerical references within the same paragraph is noted. It is advisable to choose one consistent citation style throughout the manuscript to maintain uniformity and readability. I recommend standardizing to APA style as it provides clearer attribution and is commonly used in the field. This change should be applied throughout the document to ensure coherence and to align with academic standards.
Line 31: The use of "in contrast" is incorrect as the relationship described between the shorter and longer fasting periods is expected and does not present a contrasting scenario.
Line 33: The statement about feasibility and satisfactory growth performance needs to be clarified with empirical data or removed if unsubstantiated.
Line 35: The focus on fish growth without discussing plant outcomes undermines the discussion on aquaponics; either adjust the title or include plant growth results.
Line 97: Verify the duration of the experiment as there is inconsistency in the manuscript regarding the number of days.
Line 108-109: Correct the reference to salinity adaptation for glasswort plants; they are naturally halophytic and do not require such adjustments.
Line 113: The volume calculations for the aquaponic systems are confusing and need clarification.
Recommendation:
I would recommend this manuscript for publication contingent upon the implementation of the suggested revisions, which would enhance its clarity, impact, and contribution to the field of aquaculture research.

Experimental design

While the experimental design is clearly outlined, there are several areas where improvements are necessary to meet the expected standards.

Validity of the findings

no comment

Additional comments

no comment

Reviewer 2 ·

Basic reporting

In the introduction I recommend introducing bibliographical references on the experience of aquaponics in salt water, the most common plant and fish species and whether there are studies about salicornia, seabass and seabream. In addition, previous studies on the starvation and refeeding of fish in aquaponics systems. Some of these studies are mentioned in the discussion, so they could be included in the introduction as well.

I recommend that in the letters of the tables indicating the Tukey test, assign the letter “a” to the highest values, b and c to the following values. The letters are usually assigned from the highest to the lowest values.

Experimental design

No comment

Validity of the findings

No comment

Additional comments

Line 74-79: I recommend moving this paragraph after the paragraph referring to seabass and seabream fish.
Line 171-72: I recommend transferring this information to the introduction: To the authors' knowledge, this is the first study investigating starvation's effect on compensatory growth and how glasswort absorbs and utilizes nutrients from fish waste.
Line 556-557: It can also be noted that glasswort is considered a valuable source of food for human consumption, in fact it is used as an ingredient for the preparation of dishes in different parts of the world, including Europe.

Reviewer 3 ·

Basic reporting

.

Experimental design

.

Validity of the findings

.

Additional comments

General comments

1. The manuscript is written clearly and concisely, and follows the rules of good English composition.

2. The subject is logically presented and developed. The flow of the paper is logical and clear, it includes sufficient details on the background importance.

3. The objectives are clear and meaningful.

4. The approach is experimental and the proposed method is validated on study.

5. Conclusions are logical, mindful and sufficient at that stage of the demonstration.

6. Tables and figures are sufficient and necessary. The review of the literature is ample and adequate.

7. The length of the presentation is appropriate in order to provide enough elements in the description of the methods.

8. The paper is well written and I only have some minor specific comments, as presented below.


Specific comments

9. line 6: “author Konstantinos Babouklis” change by Babouklis Konstantinos like the name of the other authors.

10. line 77: “… (Eroldogan et al. 2006).” add 2006a or 2006b for determine the reference exact and noted also to the References no 26 (line 655) or 27 (line 657).

11. line 108: “… for 70 days ” add the right: 75 days. Also note the initial fish density (Kg/m3).

12. line 128: “Vlahos et al. 2023” add 2006a or 2006b for determine the reference exact.

13. line 148: “… less than 5%” note in which unit (/Hour, min, day) and in which point took place (sump filter or fish rearing tank).

14. line 249: “… (Nd*FI)” add the entire name of the abbreviation FI.

15. line 274: “water quality variables… in Table 1.” describe briefly the correspondence (fish rearing tank, raft hydroponic unit,…) of the TAN, NO2, pH. Also add this information in Table 1 and determine the terms GB (line 7) and FT (line 9) at NO2 level. Same must be done in Figures 3, 4, 5.

16. line 277: “… than in treatments b” replace b by B.

17. line 313: “The Fulton coefficient factor (K) ” describe briefly the term.

18. line 355: “… studies [4,16] ” add the references, 4: Allsopp et al. 2008, 16: Castagna et al. 2022.

19. line 384: “…. rock samphire” add the scientific name.

20. line 404: “…. Gadus morhua” add the common name (optional). The same for all other species that follow in the text.

21. line 414: “…. Atlantic cod” add the scientific name.

22. line 439: “….Sturgeon, …salmon” add the scientific name.

23. line 453: “… (Eroldogan et al. 2006).” add 2006a or 2006b for determine the reference exact and noted also to the References no 26 (line 655) or 27 (line 657).

24. line 489: “…. trainbow trout” add the scientific name.

25. line 492: “Arctic char” add the scientific name.

26. line 533: “…. pacific white shrimp” add the scientific name.

27. line 533: in References, the no 43 is doubled (same with no 44).


Overall opinion

The paper is original and well written.

My overall opinion is that the paper can be accepted for publication in its general current form, after taking into account the minor clarifications comprised in the specific comments.

---

## Round 0.2 · accepted · Accept

Dear Dr. Berillis

I would like to thank you and your co-authors for making the corrections and changes requested by the reviewers. I read and checked carefully your valuable article and I am happy to inform you that your article has been accepted for publication in PeerJ.

Best regards

Reviewer 1 ·

Basic reporting

Dear Editor,

The authors have addressed all the reviewer's comments thoroughly and have made significant improvements to the manuscript. Given their comprehensive revisions and effort, I believe the manuscript is now acceptable for publication.

Best regards

Experimental design

The authors have demonstrated a clear and relevant research question that addresses a significant knowledge gap within the scope of the journal. The study has been conducted with rigorous methods and high ethical standards, and the detailed methodology ensures reproducibility.

Validity of the findings

The authors have provided robust and statistically sound data, ensuring transparency and reliability. Their conclusions are well-stated, directly linked to the original research question, and limited to the supporting results, maintaining scientific integrity.

Additional comments

This study addresses an intriguing topic and its acceptance will significantly contribute to the scientific literature.

Reviewer 2 ·

Basic reporting

no comment

Experimental design

no comment

Validity of the findings

no comment

Additional comments

no comment

Reviewer 3 ·

Basic reporting

see below

Experimental design

see below

Validity of the findings

see below

Additional comments

I would like to let you know that after checking the text, I confirm that the authors have corrected and adjusted all points of the text that were indicated/ suggested.